# Brain Proteome and Behavioural Analysis in Wild Type, BDNF^+/−^ and BDNF^−/−^ Adult Zebrafish (*Danio rerio*) Exposed to Two Different Temperatures

**DOI:** 10.3390/ijms23105606

**Published:** 2022-05-17

**Authors:** Elisa Maffioli, Elisa Angiulli, Simona Nonnis, Francesca Grassi Scalvini, Armando Negri, Gabriella Tedeschi, Ivan Arisi, Flavia Frabetti, Salvatore D’Aniello, Enrico Alleva, Carla Cioni, Mattia Toni

**Affiliations:** 1Department of Veterinary Medicine and Animal Science, Università degli Studi di Milano, Via dell’Università 6, 26900 Lodi, Italy; elisa.maffioli@unimi.it (E.M.); simona.nonnis@unimi.it (S.N.); francesca.grassiscalvini@unimi.it (F.G.S.); armando.negri@unimi.it (A.N.); gabriella.tedeschi@unimi.it (G.T.); 2Department of Biology and Biotechnology “Charles Darwin”, Sapienza University, Via Alfonso Borelli 50, 00161 Rome, Italy; elisa.angiulli@uniroma1.it (E.A.); carla.cioni@uniroma1.it (C.C.); 3CRC I-WE (Coordinating Research Centre: Innovation for Well-Being and Environment), University of Milan, 20134 Milan, Italy; 4Bioinformatics Facility, European Brain Research Institute (EBRI) “Rita Levi-Montalcini”, 00161 Rome, Italy; i.arisi@ebri.it; 5Institute of Translational Pharmacology (IFT), National Research Council (CNR), 00131 Rome, Italy; 6Department of Experimental, Diagnostic and Specialty Medicine, University of Bologna, 40136 Bologna, Italy; flavia.frabetti@unibo.it; 7Biology and Evolution of Marine Organisms, Stazione Zoologica Anton Dohrn Napoli, Villa Comunale, 80121 Napoli, Italy; salvatore.daniello@szn.it; 8Center for Behavioural Sciences and Mental Health, Istituto Superiore di Sanità, 00161 Rome, Italy; enrico.alleva@iss.it

**Keywords:** BDNF, temperature, zebrafish, proteomic, behaviour

## Abstract

Experimental evidence suggests that environmental stress conditions can alter the expression of BDNF and that the expression of this neurotrophin influences behavioural responses in mammalian models. It has been recently demonstrated that exposure to 34 °C for 21 days alters the brain proteome and behaviour in zebrafish. The aim of this work was to investigate the role of BDNF in the nervous system of adult zebrafish under control and heat treatment conditions. For this purpose, zebrafish from three different genotypes (wild type, heterozygous BDNF^+/−^ and knock out BDNF^−/−^) were kept for 21 days at 26 °C or 34 °C and then euthanized for brain molecular analyses or subjected to behavioural tests (Y-maze test, novel tank test, light and dark test, social preference test, mirror biting test) for assessing behavioural aspects such as boldness, anxiety, social preference, aggressive behaviour, interest for the novel environment and exploration. qRT-PCR analysis showed the reduction of gene expression of BDNF and its receptors after heat treatment in wild type zebrafish. Moreover, proteomic analysis and behavioural tests showed genotype- and temperature-dependent effects on brain proteome and behavioural responding. Overall, the absent expression of BDNF in KO alters (1) the brain proteome by reducing the expression of proteins involved in synapse functioning and neurotransmitter-mediated transduction; (2) the behaviour, which can be interpreted as bolder and less anxious and (3) the cellular and behavioural response to thermal treatment.

## 1. Introduction

BDNF (Brain Derived Neurotrophic Factor) is a phylogenetically well-conserved growth factor belonging to the neurotrophin family [1,2] together with nerve growth factor (NGF), neurotrophin-3 (NT-3) and neurotrophin 4/5 (NT-4/5) [3], neurotrophin-6 (NT-6) [4] and neurotrophin-7 (NT-7) [5]. BDNF exerts its function by binding to two different receptors, TrkB [6,7] and p-75NTR [8,9], and it plays multiple and critical roles both during embryonic development and adult life. BDNF is implicated in the survival, maintenance and growth of central and peripheral neurons, the maturation of neuronal circuitry and synaptic plasticity [10]. Moreover, BDNF is involved in long-term potentiation (LTP) in the hippocampus [11,12,13], and the decreased expression of BDNF contributes to hippocampal atrophy and neuronal loss in experimental animals [14]. Furthermore, BDNF is involved in ocular dominance plasticity in the visual cortex [15,16]. BDNF dysfunctions may play a role in the pathophysiology of psychiatric and neurological disorders such as Huntington’s and Alzheimer’s diseases, depression, suicidal behaviour, schizophrenia and dementia [17,18,19,20,21,22].

The literature data suggest that stress conditions may affect BDNF expression influencing cognitive abilities and animal behaviour. Studies on mice and rats showed that stress conditions such as acute and chronic immobilization, maternal deprivation, foot shock and social stress cause a reduction of BDNF gene [23,24,25,26,27,28,29] and protein [13,29,30] expression in the brain.

The reduced expression of BDNF has been associated with alterations in animal behaviour. Indeed, the decreased expression of BDNF determines rearrangements in brain areas controlling emotional reaction and influences social behaviour and exploration in rats [31]. Moreover, the reduced expression of BDNF in the hippocampus and prefrontal cortex due to prenatal stress compromises social behaviour of periadolescent rats [32]. Low BDNF expression reduces cognitive functioning and induces social impairment in both animals and humans [33,34]. Furthermore, BDNF deficiency due to anxiety-related behaviour in mice increases aggressive tendencies and anxiety levels in the open field and elevated-plus maze assays [33]. Other reports show that the reduction of BDNF expression in Fmr1 knockout mice worsens contextual fear learning and spatial learning and improves hyperactivity and sensorimotor deficits in mice under stress conditions [35].

Temperature variation is a potential stress factor for all organisms that can affect their physiology, behaviour and welfare, especially for ectothermic animals whose body temperature depends on the ambient temperature, such as fish [36,37,38,39]. No data are available on the effect of temperature on BDNF expression. The zebrafish is a poikilotherm and eurytherm cyprinid which is widely used as a model organism in several research fields, such as developmental biology, physiology, neuroscience and ecotoxicology, and it is a good model for the study of BDNF [1,40,41].

Recent studies have shown that zebrafish can be useful also for studying the effect of thermal variation as it is characterised by wide thermal tolerance from 6.7 to 41.7 °C [42,43]. This biological condition allows the animal to live in freshwater streams of the South East Asian regions that are characterised by daily temperature fluctuations of ∼ 5 °C [44] and wide seasonal variations from 6 °C in winter to more than 38 °C in summer [45]. Such physiological features enable testing the effects of large temperature variations within the animal’s tolerance polygon [44].

It has been shown that the exposure of adult zebrafish to low (18 °C) and high (34 °C) temperatures, compared to 26 °C, alters both gene expression in the liver and the lipid content in gonads and muscle [46]. Moreover, studies carried out by our research group showed that temperature variations alter brain protein expression and behaviour in both acute (4 days) and chronic (14–28 days) treatments [47,48,49].

The aim of this work was to investigate the role of BDNF in the nervous system by acquiring information on its involvement in neurochemical and behavioural responses both under control conditions and following heat treatment in order to understand how the variation of BDNF expression possibly altered the response to temperature change described in our previous studies on the subject [47,48]. For this purpose, adult zebrafish from three different genotypes were used: wild type (WT), heterozygous BDNF^+/−^ (HT) and knock out BDNF^−/−^ (KO). The KO mutant line was recently produced by our research group [50] and constituted a new tool for the study of the complex role of BDNF, which supports and completes the mouse model. Unlike BDNF^−/−^ mice which are characterised by early postnatal lethality [51], KO zebrafish are viable in the adult stage, capable of producing fertile and stable offspring from generation to generation. Therefore, this model allows for investigating of the role of BDNF in all stages of development of the organism, from the embryo to the larvae [50] and finally to the adult. Overall, the results here presented suggest that environmental temperature influences metabolic and biochemical parameters, which in turn can alter the behaviour of the animals.

## 2. Results

### 2.1. Chronic Exposure to High Temperatures Reduces the Gene Expression of BDNF and Its Receptors in the Brain of Adult Zebrafish

A pilot experiment was performed to analyse the effect of chronic exposure to elevated temperatures on the gene expression of BDNF and its receptors. The changes in brain mRNA levels were examined by qRT-PCR in WT adult zebrafish kept at 26 °C (control temperature) or 34 °C (high temperature) for 21 days. The results showed that the heat treatment reduces the gene expression of BDNF, Trk2a, Trk2b and p75 (Figure 1), suggesting a strong down-regulation of BDNF signalling. This data is consistent with results obtained in mammals showing that environmental stress conditions reduce BDNF gene expression in some brain areas [23,29] and suggest that BDNF is involved in the response to thermal changes in zebrafish.

In mammalian models, the reduction of BDNF expression has been associated with behavioural alterations [31,32,33,34,35], and previous work demonstrated brain proteome and behaviour alteration in adult zebrafish kept at 34 °C for 21 days compared to controls at 26 °C [47,49]. Consequently, it can be hypothesised that the down-regulation of the BDNF pathway resulting from heat treatment is involved in the behavioural response of zebrafish.

### 2.2. Genotype and Temperature Affect the Brain Proteome and Behaviour of Adult Zebrafish

To further investigate the role of BDNF in the nervous system and its role in the response to heat treatment, the main experiment was performed keeping wild type (WT), heterozygous BDNF^+/−^ (HT) and knock out BDNF^−/−^ (KO) adult zebrafish at 26 °C or 34 °C for 21 days. At the end of the heat treatment, some subjects were sacrificed to perform brain proteome analysis and others were subjected to behavioural tests to evaluate the genotype effect and temperature effect.

#### 2.2.1. Proteomic Analysis

The brain proteome analysis was performed by a quantitative shotgun label-free strategy (Figure 2). The proteomic analysis allowed identifying proteins common to all data sets as well as proteins differentially expressed in the different conditions as reported in the Venn diagrams of the comparison KO26 vs. WT26, HT26 vs. WT26, KO26 vs. HT26, KO34 vs. WT34, HT34 vs. WT34, KO34 vs. HT34, KO34 vs. KO26 and HT34 vs. HT26 (Appendix A). ClueGo (Cytoskape 3.9.1) (Figure 3 and Figure 4, Appendix A) and Panther (Figure 5 and Figure 6, Appendix A) analyses were performed to detect genotype and temperature effects. The list of the proteins differentially expressed in the comparisons is reported in Appendix A.

In HT26 vs. WT26, HT showed a reduced expression of proteins involved in key cellular events such as transcription, mRNA processing, splicing, mRNA transport and translation, suggesting a strong influence of BDNF on protein expression (Appendix A). Furthermore, the reduced expression of the protein associated with mitochondrial fatty acid beta-oxidation and oxidative phosphorylation was observed together with glutathione metabolic processes, insulin receptor recycling and EPH-Ephrin signalling, which may be related to a decrease in the innate immune system [52]. Moreover, a decrease in the response to hypoxia and oxidative stress, which can make fish more vulnerable to environmental stressors and an increase in transmembrane transporter activity of both anions and protons (for ATP synthesis) were observed.

In KO26 vs. WT26, KO showed a reduced expression of proteins involved at various levels of cell biology such as molecular chaperones and proteasome, amino acids metabolism, energy metabolism (glycolysis, Krebs cycle, oxidative phosphorylation, pentose phosphate pathway), degradation of fatty acids, endocytosis, vesicle transport and exocytosis, and cell cycle (Appendix A). Moreover, the altered expression of proteins involved in cytoskeleton organisation and neuronal projections was observed together with the reduction of cell junction proteins. Interestingly, the knocking of BDNF reduced the expression of proteins belonging to several signalling pathways (serotoninergic, adrenergic, dopaminergic, GABAergic, glutamatergic, histaminergic, cholinergic, opioid, WNT, calcium, EGF and FGF, gonadotropin-releasing factor), suggesting a significant alteration in brain neurochemistry.

In KO26vsHT26, the knocking of BDNF was associated with the down-regulation of proteins related to the proteasome, pyruvate metabolism and Krebs cycle, vesicle transport, exocytosis, cell cycle and neurotransmitter related signal pathways which were not observed in HT (Appendix A). This suggests that the absence of BDNF has a stronger impact on cerebral neurochemistry than its partial expression.

In HT34 vs. WT34, the scenario was almost reversed in comparison to HT26vsWT26. Semaphorin receptor activity, RNA processing and translation increased, while the Krebs cycle decreased (Figure 6 and Appendix A). On the contrary, KO34 vs. WT34 (Appendix A) and KO34 vs. HT34 (Appendix A) broadly confirmed the alteration of protein expression observed in KO26 vs. WT26 showing a similar genotype effect at both 26 °C and 34 °C.

In all three genotypes, the exposure to 34 °C led to an increased expression of ribosomal proteins and a reduction in mitochondrial proteins involved in the Krebs cycle, oxidative phosphorylation, and degradation of fatty acids, as shown by common alteration in WT34 vs. WT26, HT34 vs. HT26 and KO34 vs. KO26. These results demonstrated that heat treatment leads to a reduction in the expression of mitochondrial proteins related to energy metabolism not only on WT, as previously demonstrated [47] but also on HT and KO.

At the same time, the heat treatment also resulted in a different alteration of protein expression in HT, KO and WT. KO34 vs. KO26 and HT34 vs. HT26 showed a reduction of proteins associated with amino acid metabolism and mitochondrial proteins associated with organelle development and cellular respiration, and an increase in protein associated with mRNA processing and transport was not observed in WT34 vs. WT26 (Appendix A). Moreover, opposite alteration of the expression of proteins associated with chromatin, glycolysis, and neurotransmitter-mediated transduction was observed in HT and KO as emerged from HT34 vs. HT26 and KO34 vs. KO26 (Figure 6 and Appendix A). These results suggest that the amount of BDNF may affect cellular responses to the heat treatment.

To better discriminate between temperature and genotype effect, all the data sets were further analysed using R-Bioconductor: LFQ data were normalised to the median and analysed for differential expression by the limma package, choosing |Log2FC| > 0.585 and FDR < 0.05 as differential thresholds for the limma test. The enrichment bioinformatic analysis of the proteins most differentiated, comparing each data set with all the others (reported in Appendix A), was performed by ClueGo and Panther as reported in Figure 7 (ClueGo analysis) and in Appendix A (ClueGo analysis) and Appendix A (Panther analysis). The results confirmed the observation described in the previous comparisons and highlighted the differences specific to genotype and temperature.

The genotype affects the mRNA metabolic processes, neurotransmitter secretion, actin filament polymerisation and organelle organisation. This effect is more evident in KO while HT is more like WT (Figure 7, Appendix A), suggesting that the partial expression of BDNF in HT is sufficient to prevent the alteration of the aforementioned processes. Furthermore, the reduced or lack of BDNF expression alters the electron transport chain and oxidative phosphorylation in both HT and KO, suggesting that the right amounts of neurotrophin are needed for proper mitochondrial functioning and ATP production.

On the other hand, the temperature increase at 34 °C results in the enrichment of proteins involved in the cellular response to heat, as expected, but also an alteration of proteins involved in oxidative phosphorylation independently of the genotype. Moreover, alteration in protein expression involved in pyruvate metabolism, pentose phosphate pathway, vesicle activity, transport, exocytosis, synaptic trafficking and phagosome can be observed both in WT and HT (Figure 7, Appendix A), confirming our previous results [47,48].

#### 2.2.2. Behavioural Analysis

##### Y Maze Test

The YMT was applied to evaluate the locomotor activity (Figure 8) and cognitive abilities related to the response to novelty and spatial orientation (Figure 9).

The locomotor and swimming activities of zebrafish were analysed in the YMT since the shallow depth of the water, which reduces the vertical movement (z-axis) of the fish, and the camera placed above the maze make this test more suitable for a precise analysis compared to NTT.

Genotype-dependent effects were detected by a two-way ANOVA analysis followed by Bonferroni’s post hoc test in all parameters considered (Figure 8a,d,g,j,m and refers to Appendix A for *p* value of all comparisons). At 26 °C, the time of immobility decreased in HT26 vs. WT26 and KO26 vs. WT26 (*p* < 0.0001; Figure 8c), while no differences were observed in KO26 vs. HT26. Average speed (*p* < 0.0001; Figure 8f) and distance travelled (*p* < 0.0001; Figure 8l) increased, and meandering (*p* < 0.0001; Figure 8o) decreased in KO26 vs. WT26 and KO26 vs. HT26. At 34 °C, the average speed (*p* = 0.0060 and *p* = 0.0008, respectively; Figure 8f) and distance travelled (*p* = 0.0067 and *p* = 0.0008; Figure 8l) increased while meandering (*p* < 0.0001; Figure 8o) decreased in KO34 vs. WT34 and KO34 vs. HT34.

Temperature-dependent effects were detected in all parameters considered except for meandering (Figure 8b,e,h,k,n and Appendix A). Reduced duration of immobility events in WT34 vs. WT26 (*p* < 0.00001; Figure 8c) and increased average speed (*p* = 0.0001; Figure 8f) and total distance travelled (*p* = 0.0001; Figure 8l) in WT34 vs. WT26 were observed.

The tendency of the animals to explore the new environment was evaluated by analysing the exploration of the N arm. Results showed a progressive reduction of entries in the N arm from T1 to T4 in WT at 26 °C (*p* = 0.0143 in T4 versus T1; Figure 9a, Appendix A), while no significant changes were observed in HT and KO from T1 to T4 at the same temperature (Figure 9b,c). Moreover, WT at 26 °C showed a progressive reduction of the time spent in the N arm from T1 to T4 (*p* = 0.0452 in T4 versus T1; Figure 9g), while no significant differences were observed in HT e KO (Figure 9h,i). WT at 34 °C showed no significant differences in the number of entries and in the time spent in the N arm from T1 to T4 (Figure 9d,j), confirming previous results [47,48] showing a variation in exploratory behaviour caused by the heat treatment attributable to an alteration of the cognitive abilities of the zebrafish. Moreover, no significant difference in the two parameters analysed was found in HT and KO at 34 °C (Figure 9e,f,k,l).

##### Novel Tank Diving Test

The NTT was applied to analyse the vertical exploration of the specimens in a new environment and to evaluate the response to anxiety evoked by novelty.

Genotype-dependent effects were detected by a two-way ANOVA analysis followed by Bonferroni’s post hoc test in all the parameters considered (Figure 10a,d,g,k,o and refers to Appendix A for *p* value of all comparisons). The increase in the number of transitions among bottom, middle and top areas (*p* < 0.0001, *p* = 0.0031, *p* = 0.0031 and *p* < 0.0001 respectively; Figure 10c), and the number of entries (*p* < 0.0001, *p* = 0.0018, *p* = 0.0039, *p* < 0.0001 respectively; Figure 10f), the time spent (*p* = 0.0001, *p* = 0.0005, *p* = 0.0020 and *p* < 0.0001 respectively; Figure 10i,j) and the distance travelled within the top area (*p* < 0.0001, *p* = 0.0003, *p* = 0.0025 and *p* = 0.0001 respectively; Figure 10m,n) was observed in KO26 vs. WT26, KO26 vs. HT26, KO34 vs. WT34 and KO34 vs. HT34. Whereas a lower latency to enter the top area was observed in KO26 vs. WT26 and KO26 vs. HT26 (*p* < 0.0001 and *p* = 0.0032, respectively; Figure 10q).

Temperature-dependent effects were detected in all parameters considered (Figure 10b,e,h,l,p. and refers to Appendix A for *p* value of all comparisons). The increase in the number of transitions among bottom, middle and top areas (*p* = 0.0191; Figure 10c) and the distance travelled within the top area (*p* = 0.0309; Figure 10m), and lower latency to enter the top area (*p* < 0.0001; Figure 10q) were observed in WT34 vs. WT26. A reduction in the latency to enter the top area (*p* = 0.0027; Figure 10q) was observed in HT34 vs. HT26, while no difference was detected in KO34 vs KO26 in all parameters assessed.

##### Light and Dark Preference Test

LDT measures the natural preference of adult zebrafish for a dark environment (scototaxis), and it is used to assess fish exploratory activity and anxiety-like behaviour [53].

Genotype-dependent effects were detected by a two-way ANOVA analysis followed by Bonferroni’s post hoc test in the time spent in the dark area (Figure 11a and refers to Appendix A for *p* value of all comparisons). The time spent in the dark area decreased in HT26 vs. WT26 and KO26 vs. WT26 (*p* = 0.0017 and *p* < 0.0001, respectively; Figure 11c), while no significant difference was detected at 34 °C.

Temperature-dependent effects were detected in time spent in a dark area (Figure 11b and Appendix A) with a reduction in WT34 vs. WT26 (*p* = 0.0006; Figure 11c). On the contrary, no significative variation was observed in HT34 vs. HT26 and KO34 vs. KO26 (Figure 11c), demonstrating that heat treatment in HT and KO does not further alter scototaxis. Moreover, the number of transitions between light and dark decreased in HT34 vs. HT26 (*p* = 0.0123; Figure 11f) and increased in KO34 vs. HT34 (*p* = 0.0092; Figure 11f).

##### Social Preference Test

The SPT is based on the social nature of zebrafish [54] that exhibit group behaviours like shoaling and schooling [55,56].

Genotype-dependent effects were detected by a two-way ANOVA analysis followed by Bonferroni’s post hoc test only in transition to the social area (Figure 12a,d,g and refers to Appendix A for *p* value of all comparisons). Increased transition to social area was observed in KO34 vs. WT34 (*p* = 0.0020) and KO34 vs. HT34 (*p* = 0.0002; Figure 12c).

Temperature-dependent effects were observed in time spent and distance travelled in the social area (Figure 12e,h and Appendix A). The reduction of the time spent (*p* = 0.0018; Figure 12f) and the distance travelled in the social area (*p* = 0.0076; Figure 12i) was observed in WT34 vs. WT26, confirming a reduced interest in conspecifics at high temperatures [49].

##### Mirror Biting Test

The MBT is a paradigm used for studying social and aggressive behaviour in zebrafish based on mirror-image stimulation [57].

Genotype-dependent effects were detected by a two-way ANOVA analysis followed by Bonferroni’s post hoc test in mirror bites (Figure 13g) and mirror biting latency (Figure 13j; refers to Appendix A for *p* value of all comparisons). The number of mirror bites decreased in HT26 vs. WT26 (*p* = 0.0057; Figure 13i) and KO26 vs. WT26 (*p* < 0.0001; Figure 13i) and the mirror biting latency increased in KO26 vs. WT26 (0.0076; Figure 13l).

Temperature-dependent effects were detected in the mirror approach zone entries (Figure 13b) and in mirror bites (Figure 13h; refers to Appendix A for *p* value of all comparisons). The increase of mirror approach zone entries in HT34 vs. HT26 (*p* = 0.0001; Figure 13c) and the reduction of mirror bites in WT34 vs. WT26 (*p* = 0.0361; Figure 13i) were observed.

## 3. Discussion

Our previous studies have shown that exposure of adult zebrafish to 34 °C for 21 days alters brain proteome and behaviour [47]. The present study demonstrates that the same conditions reduce the gene expression of BDNF and its receptors, suggesting that some of the previously observed variations may be related to the reduced BDNF expression. To further investigate the involvement of BDNF in the regulation of brain protein expression and behaviour, adult zebrafish of three different genotypes (WT, HT, and KO) were maintained at temperatures of 26 °C or 34 °C for 21 days and subjected to brain proteome and behavioural analysis. Results allowed us to evaluate genotype- and temperature-dependent effects and if the variation in BDNF expression alters the response to temperature change.

In proteomic analysis, the genotype effect consisted of altered expression of brain proteins in HT and KO zebrafish. The expression of proteins involved in transcription, translation, protein folding and degradation is altered in KO compared to WT at 26 °C. The lack of BDNF expression is associated with the alteration of the actin filaments polymerization, neurotransmitters secretion and mRNA metabolic processes in keeping with the ability of BDNF to activate the translation machinery and protein synthesis in mammalian neurons [58,59,60,61,62,63,64,65]. Proteins involved in important cellular events are down-regulated in HT26 vs. WT26 and KO26 vs. WT26. The altered expression of proteins involved in oxidative phosphorylation detected both in HT and KO at 26 °C or 34 °C is consistent with the involvement of BDNF in the regulation of energy homeostasis by increasing glucose transport [66], mitochondrial biogenesis [67] and efficiency [68,69], as observed in mammals. Moreover, the down-regulation of proteins associated with neuronal vesicles found in KO zebrafish is in line with the dramatic reduction in the number of synapses and vesicle sprouting caused by the reduced expression of BDNF or TrkB in mammalian models [70,71,72,73,74,75,76,77,78].

The down-regulation of proteins associated with neurotransmitter-mediated signal transduction in KO zebrafish could be explained by the reduction of synaptic vesicle exocytosis, and it is consistent with the role of BDNF in the development and functioning of neurotransmitter systems described in mammals. For example, the down-regulation of the serotonin pathway is coherent with the facilitating effect of BDNF on the brain 5-HT system [79]. In fact, it has been reported that BDNF^+/−^ mice result in an impairment of forebrain 5-HT levels [80,81,82] and that BDNF increased the number of 5-HT expressing cells [83,84,85,86] and the sprouting of 5-HT axons in mammals [87,88]. Moreover, the down-regulation of the histamine and calcium transduction pathway is consistent with evidence demonstrating respectively the increase of histamine turnover in rat hypothalamus after BDNF treatment [89] and BDNF ability to induce Ca^2+^ influx in mammals [69]. Finally, the reduction of enkephalin-associated proteins is in line with the degeneration of enkephalinergic striatal projection neurons caused by insufficient levels of BDNF in mutant mice and with the recovery of neuronal dysfunctions after BDNF administration [17].

The reduction in cell cycle-associated proteins observed in KO zebrafish is consistent with the role of BDNF in cell cycle control [90] observed in mammals in which the knockdown of BDNF reduced cell growth and proliferation [91] and knockdown of TrkB expression induced G0/G1 arrest and suppressed proliferation activity [92].

Temperature is an abiotic parameter critical for animal life as it influences enzymatic activities and cell biology. As expected [47,48], the thermal variation alters the cerebral proteome and determines alterations of the protein expression in the three genotypes. The alteration of cellular response to heat, glycolysis/gluconeogenesis, pentose phosphate pathway, phagosome, pyruvate metabolism and synaptic vesicle endocytosis occurs in both WT and HT at 34 °C. These results confirm the strong influence of temperature on zebrafish energy metabolism already observed in our previous works [47] and suggest that the resulting low bioavailability of ATP may strongly affect neuronal cell biology. In KO at 34 °C, the alteration of proteins involved in DNA recombination, nucleosome assembly and chromatin silencing, mRNA metabolic process, organelle organisation, mitochondrial electron transport, and actin filament polymerisation is observed. Moreover, alteration of neurotransmitter secretion is observed in both HT and KO. Interestingly, the effect of temperature increase appears more similar in WT and HT than in KO.

This confirms the key role that BDNF plays at the neuronal level but shows clear differences between KO cells that do not express BDNF and HT cells characterised by a partial expression of neurotrophin.

In fact, the proteomic data suggest that the complete absence of BDNF strongly alters the cellular capacity to cope with the increase in temperature, while the partial expression of neurotrophin may be sufficient to allow a cellular response more like that of WT.

The NTT, LDT, SPT, MBT and YMT behavioural tests on WT, HT and KO zebrafish maintained under the same conditions (26 °C or 34 °C for 21 days) allowed us to investigate whether neurochemical alterations due to genotype and temperature affected zebrafish behaviour. These tests have already been validated on zebrafish and allowed the identification of behavioural variations following specific treatments [48,49]. Overall, present results showed both temperature- and genotype-dependent effects on zebrafish behaviour (Figure 14).

The analysis of swimming activity in the YMT revealed both a temperature- and a genotype-dependent effect on zebrafish hyperactivity. Total distance travelled, average speed increase and immobility time (freezing) were all reduced in WT34 vs. WT26 and KO26 vs. WT26. Moreover, the meandering behaviour was reduced at 26 °C in KO compared to WT. Although the observed differences in swimming speed could be attributed to different performances at the muscular level (greater efficiency of muscle contraction or transmission at the level of the motor endplate), the meandering reduction suggests a different strategy of environment exploration that is not dependent on muscular performance. In fact, the meandering assessed as the ratio between the absolute turn angle and the total distance travelled indicates a movement without a fixed direction or path [93], and low values correspond to a swimming behaviour characterised by straight trajectories. These results are consistent with experimental evidence showing that the down-regulation of the BDNF pathway in mammals is associated with freezing reduction [94,95,96,97,98,99], locomotor activity increase [98] and highly linear movement [98].

The analysis of Y-maze exploration during the four trials (T1–T4) confirmed the ability of WT at 26 °C to modulate the explorative behaviour by progressively reducing their interest in the N arm based on previous experience [47,48] and demonstrated that both temperature and genotype can alter zebrafish exploration patterns. In fact, no statistically significant differences were observed from T1 to T4 in the number of entries and time spent in all other conditions assessed.

The environmental exploration was further analysed by NTT and LDT, two behavioural tests based respectively on geotaxis and scototaxis [100]. The NTT is the analogue of the open field test in rodents and is based on the innate-type escape “diving” behaviour of zebrafish in novel environments, whereas the LDT is based on the innate stereotyped tendency of adult zebrafish to prefer dark regions with respect to the light areas, a species-specific antipredatory response. Both genotype- and temperature-dependent effects on geotaxis and scototaxis were observed. In fact, WT at 34 °C and KO at 26 °C showed a greater exploration of the top area of the tank in the NTT and of the light area in the LDT compared to WT at 26 °C. These results confirm the effect of the heat treatment observed previously on WT [48,49] while demonstrating that the down-regulated expression of BDNF alters zebrafish scototaxis and geotaxis. These results are inconsistent with the reduction in the time spent in the light area [101,102,103,104,105] in the LDT and reduced exploration of the central area in the open field test [101,102,103,104,105] observed in rodents with a reduced expression of BDNF suggesting different effects in fish and mammals. The bolder and risk-taking behaviour exhibited by WT zebrafish at 34 °C and by KO at 26 °C would, in fact, potentially expose the animal to major predation in their natural environment.

The MBT was then used to investigate further the apparent boldness of the zebrafish. This test is based on the reaction stimulated by the image reflected in a mirror, to which the fish can respond aggressively by biting it. A reduction in the number of bites was observed in WT34 vs. WT26, HT26 vs. WT26 and KO26 vs. WT26 suggesting that heat treatment or reduced/absent expression of BDNF compromise the intraspecific aggressive behaviour of fish. These results are consistent with experimental evidence in mammals showing the association between increased brain BDNF level and aggressive episodes [106,107,108]. However, literature data on the relationship between BDNF and aggression in mammals are conflicting as other studies showed increased aggressive behaviour in heterozygous BDNF^+/−^ and conditional knockout rodent models [80,96,109,110].

In the SPT, a temperature-dependent effect characterised by a reduction in time spent and distance travelled in the social area in WT34 vs. WT26 was observed, confirming previous results [49], whereas no genotype effect was detected.

Overall, present results show that both heat treatment and genotype strongly influence zebrafish brain protein expression and behaviour. Both high-temperature exposure and BDNF reduced expression apparently exert an anxiolytic-like effect on fish behaviour, considering that the reduction of freezing events [111], meandering values [93], time spent at the bottom of the tank [111,112,113,114] and time spent in the dark compartment [53] are interpreted as anti-anxiety effects. However, the conspicuous alteration of the brain proteome could result in an altered perception or processing of external stimuli resulting in a bolder behaviour that could be misinterpreted as reduced anxiety. Thus, the increase in straight and non-exploratory swimming in YMT, the greater exploration of top and light areas respectively in NTT and LDT, together with the reduction of bites in MBT and the reduced modulation of environmental exploration in YMT, may correspond more to zebrafish inability to process some characteristics of the environment and recognise them as dangerous than to its enhanced boldness and reduced anxiety. Moreover, results suggest that BDNF expression levels in the zebrafish brain play a significative role in response to temperature increases at both cellular and behavioural levels and that the reduced or absent BDNF expression may have different effects. In fact, the exposure to 34°C determines an alteration of protein expression and behaviour in KO fish different from that observed in WT and HT at the same temperature.

## 4. Materials and Methods

### 4.1. Ethical Note

Animal husbandry and experimental procedures were performed in accordance with the guidelines approved by the Animal Care Committee and authorised by the Italian Ministry of Health (protocol number 290/2017-PR) and in accordance with the European directive 2010/63 on the protection of animals used for scientific purposes. The health status and the well-being of all animals involved in the study were checked daily for the duration of the heat treatment and the subsequent behavioural tests. All efforts were undertaken to reduce psychophysical animal suffering, and no experimental subject died during the experimental procedures (fish housing, adaptation period, heat treatment and behavioural testing).

### 4.2. Subjects

A total of 252 adult (12 months old) AB zebrafish (50:50 male:female) belonging to three genotypes (100 WT, 76 HT BDNF^+/−^ and 76 KO BDNF^−/−^) were used in the present study. BDNF^+/−^ and BDNF^−/−^ mutant fish were produced by CRISPR/Cas9 genome editing system and genotyped as previously described [50,115]. According to standard procedures, the zebrafish were initially housed at 26 ± 1 °C in a stand-alone rack system with transparent polycarbonate tanks and a dedicated water supply [116] in the zebrafish facility of the University of Bologna (Italy). The zebrafish were raised and manipulated equally under a 14/10-h light/dark photoperiod (light 6 am–8 pm). The number of experimental subjects was chosen, given the reduction criteria, on the basis of data in the literature suggesting that significant data may be obtained with *n* = 12–15 per group for strong effects and with *n* = 20–25 per group to detect less evident effects [117].

### 4.3. Experimental Setting

The experimental setting (room, equipment, tanks and conditions adopted) was the same used in our previous works [47,48,49], to which refer for further details.

Overall, 12 identical tanks (W40 × D30 × H30 cm) (home-tanks) were equipped with digital thermostats (Eden 430, Hörstel, Germany) connected to a heating coil (Eden 415, 230 V, 50/60 Hz, 80 W, Hörstel, Germany), with external filter systems (Eden 511 h, Hörstel, Germany) and with aerator for aquaria (SicceAIRlight, 3300 cc/min 200 L/h, Vicenza, Italy). The interior enrichment of each tank (consisting of a heating coil, inlet and outlet pipes of the filters and aerator) was replicated identically in all the tanks.

The chemical/physical characteristics of tank water were checked at least two times per week by the Sera aqua-test box kit (Sera Italia srl, Bologna, Italy). Faeces and the remaining food waste were removed from the animal tanks at least three times per week. During the tank-cleaning operations, a water exchange of about 20–30% per week was performed to restore the correct volume of water and to maintain its chemical-physical parameters.

Zebrafish were fed three times a day (10 am, 2 pm and 6 pm) with commercial dry granular food (TropiGranMIX, Dajanapet, Bohuňovice, Czech Republic) by using automatic fish feeders (Eden 90, Eden Water paradise, Hörstel, Germany), allowing zebrafish to feed themselves according to their appetite for the entire duration of the experiment.

Adaptation period. At the beginning of the experiments, zebrafish were transferred to the home tank and maintained at the density of 1 zebrafish/L at 26 ± 1°C (control temperature) for 10 days to acclimate to the tank.

Heat treatment. The water temperature in the treatment home tanks was gradually brought from 26 °C to 34 °C in 72 h. Fish were then maintained at the two temperatures of 26 ± 1 °C (control) and 34 ± 1 °C (treatment) for 21 days. The two temperature values were chosen according to [46] within the zebrafish vital range and corresponded to temperatures the fish cope with within the natural environment. During adaptation and thermal treatment, the behaviour of fish in each tank was observed every day for 5 min by two operators (M.T. and E.A.) to evaluate physical indicators for fish welfare [47]. The zebrafish used both in the pilot and the main experiment were subjected to an adaptation period and heat treatment.

### 4.4. Pilot Experiment

A total of 24 WT zebrafish were divided into 2 home tanks for the pilot experiment aimed at analysing the gene expression of BDNF and its receptors. After the adaptation period, one tank was randomly chosen as control (26 °C) and one as treatment (34 °C). At the end of the thermal treatment, zebrafish were euthanised individually by prolonged immersion in a solution of the anaesthetic tricaine methane sulfonate MS-222 (300 mg/L). The brain was removed by surgical dissection and stored in RNAlater (Thermo Fisher, Waltham, MA, USA) at −80 °C for subsequent analysis. Total RNA was isolated from whole brain tissue with the use of PureLink RNA^®^ Mini Kit (Thermo Fisher, Waltham, MA, USA) according to the manufacturer’s instructions and quantified spectrophotometrically by Optizen Pop Bio (Mecasys, Daejeon, Republic of Korea). The mRNAs obtained were reverse-transcribed into cDNAs using oligo dT and SuperScript™ II Reverse Transcriptase (Thermo Fisher, Waltham, MA, USA); then, cDNA was stored at −20 °C until use. The qPCR was performed in 10 µL with a primer concentration of 1 μM, 10 ng cDNA and 1× SYBR Green Qpcr Master Mix (EURx, Gdansk, Polska) and conducted in the Line-Gene K PCR (BIOER, Hangzhou, China). The amplification setup consisted of an initial denaturation step at 95 °C for 2 min and 40 cycles of denaturation at 95 °C for 5 s, annealing at 64 °C for 30 s and extension at 72 °C for 30 s. In a preliminary test, the expression of four housekeeping genes (β-actin, elongation factor 1α, ribosomal protein L13 and α-tubulin) was analysed at 26 °C and 34 °C to verify the stability of their expression at the two temperatures. All housekeeping genes showed good and similar stability [118], and we chose tubulin for the higher reaction efficiency (the lowest CT values, Appendix A). A total of three replicate samples were analysed in triplicates on separate reactions to avoid technical measurement errors. Primer pairs used for qPCR analyses were designed by using the Primer3 software (version 4.1.0 http://https://primer3.ut.ee/ (accessed on 11 December 2018)) [119]. Primer sequences are reported in Appendix A. The relative expression levels for each gene were calculated by the 2^−ΔΔCT^ method and normalised using the relative expression of α-tubulin.

### 4.5. Main Experiment

In this study, 76 WT, 76 HT and 76 KO zebrafish were used. The 76 individuals of each genotype were divided into 4 home tanks (19 fish per tank), of which two were randomly chosen as control (26 °C) and two as treatment (34 °C). Of the 76 zebrafish, 8 individuals (4 at 26 °C and 4 at 34 °C) were euthanised for brain proteome analysis, 28 individuals (14 individuals at 26 °C and 14 at 34 °C) were tested in the Y-maze test (YMT) and subsequently in the mirror biting test (MBT), and 40 individuals (20 at 26 °C and 20 at 34 °C) in the novel tank test (NTT), light and dark test (LDT), social preference test (SPT).

#### 4.5.1. Shotgun Mass Spectrometry Analysis for Label-Free Proteomics

The brain proteome was analysed by a shotgun label-free proteomic approach to identify and quantify expressed proteins [120]. Four whole brains for each genotype and temperature were homogenised, reduced, alkylated and digested according to [48]. LC-ESI-MS/MS analysis was performed on a DionexUltiMate 3000 HPLC System with a PicoFritProteoPrep C18 column (200 mm, internal diameter 75 μm). Gradient: 1% ACN in 0.1% formic acid for 10 min, 1–4% ACN in 0.1% formic acid for 6 min, 4–30% ACN in 0.1% formic acid for 147 min and 30–50% ACN in 0.1% formic for 3 min at a flow rate of 0.3 μL/min. The eluate was electro-sprayed into an LTQ Orbitrap Velos (Thermo Fisher Scientific, Bremen, Germany) through a Proxeon nanoelectrospray ion source. The MS was operated in a positive mode in data-dependent acquisition mode to automatically alternate between a full scan (m/z 350–2000) in the Orbitrap (at resolution 60,000, AGC target 1,000,000) and subsequent CID MS/MS in the linear ion trap of the 20 most intense peaks from full scan (normalised collision energy of 35%, 10 ms activation). Data acquisition was controlled by Xcalibur 2.0 and Tune 2.4 software (Thermo Fisher Scientific, Bremen, Germany). Dynamic exclusion was set to 60 s. Rejection of +1 and unassigned charge states were enabled [121].

A database search was conducted against the *Danio rerio* Uniprot sequence database (https://www.uniprot.org/proteomes (accessed on personal PC), release 11 December 2018) with MaxQuant (version 1.6.0.1) (Max Plank Institute of Biochemistry, Munchen, Germany) software. The initial maximum allowed mass deviation was set to 10 ppm for monoisotopic precursor ions and 0.5 Da for MS/MS peaks. Enzyme specificity was set to trypsin, defined as C-terminal to Arg and Lys excluding Pro, and a maximum of two missed cleavages were allowed. Carbamidomethylcysteine was set as a fixed modification, while N-terminal acetylation, Met oxidation, Asn/Gln deamidation and Ser/Thr/Tyr phosphorylation were set as variable modifications.

The proteomic data were quantified and normalised by the Max Quant software using the built-in label-free quantification algorithms (LFQ) based on extracted ion intensity of precursor ions according to [122]. False protein identifications (1%) were estimated by searching MS/MS spectra against the corresponding reversed-sequence (decoy) database. Statistical analysis was performed using the Perseus software (Max Plank Institute of Biochemistry, Munchen, Germany) (version 1.6.14.0). Only the proteins present and quantified in at least 75% of the repeats were positively identified in a sample and used for statistical analysis. When focusing on specific comparisons, proteins were considered differentially expressed if they were present only in one condition or showed significant *t*-test difference (Welch’s test *p* ≤ 0.05) [120].

In order to better discriminate the effects of heat and genotype and decipher the protein and pathways deregulated by each factor, the data sets were further analysed using R-Bioconductor: LFQ data were normalised to the median and analysed for differential expression by the limma package [123] choosing |Log2FC| > 0.585 and FDR < 0.05 as differential thresholds for the limma test. The differential analysis was followed by hierarchical clustering and heatmap of samples and genes, based on the euclidean distance metric and average linkage agglomeration method, using the Log2 median normalised data.

Bioinformatic analyses were conducted by Panther software (release 16.0) [124] and ClueGo software (Cytoskape release 3.8.2) [125] to cluster enriched annotation groups of Biological Processes, Pathways, and Networks within the set of identified proteins. Functional grouping was based on Fischer’s exact test *p* ≤ 0.05 and at least 3 counts.

The mass spectrometry proteomics data have been deposited to the ProteomeXchange Consortium via the PRIDE [126] partner repository, with the dataset identifier PXD030733.

#### 4.5.2. Behavioural Tests: General Design

The behavioural tests used in this study were conducted using the same apparatus and protocol used in our previous studies [47,48,49]. The day after the end of the heat treatment (22nd day), zebrafish were subjected to YMT and MBT or NTT, LDT and SPT. These are paradigms for assessing distinct aspects of behaviour such as spatial orientation and interest in novelties, aggressive behaviour, anxiety-like state, boldness, and social preference. The LDT and NTT were performed alternately as the first or second test, and the SPT were performed as the third test in order to minimise the interference between the different tests [127,128].

As described previously [47,48,49], behavioural tests were conducted in water kept at the same temperature at which the fish had been housed (26 °C or 34 °C) to avoid acute thermal variations that could alter fish behaviour. Each individual zebrafish was captured by using a beaker and transferred from the home tank to the waiting tank (W15 × D10 × H10 cm) for 30 min until the beginning of the behavioural test [47]. The duration of each test was 10 min. At the end of each test, the zebrafish was transferred directly into the experimental tank for the next behavioural test using a beaker. After each test, the water was removed, and the apparatus was rinsed and filled with clean water. In the room, diffuse lighting was used to avoid directional lighting that could interfere with zebrafish behaviour. All tests were video recorded by a webcam (Logitech C170) that was placed one meter above (LDT, YMT) or in front of (NTT, SPT and MBT) each apparatus. The tests were conducted between 10 am and 5 pm.

##### Y Maze Test (YMT)

The Y-Maze was composed of three arms at 120 degrees to each other (W25 × D8 × H15 cm), replicating exactly the same features of the maze used successfully on zebrafish in previous works [47,48,129] (refer to [48] for graphical detailed description). A single arm was identified by the presence of geometric shapes (square, circle and triangle) that do not induce fear in the fish and for which the fish has a similar preference, as demonstrated by Cognato and collaborators [129]. The maze was filled with 4 L of water, and the water depth was 6.5 cm, enough to submerge the geometric shapes. One single task consisted of four trials (T1, T2, T3 and T4) separated by a one-hour interval. Each trial consisted of a training phase in which the fish was free to swim in the start (S) arm and in the other (O) arm for 5 min, but it could not access the novel (N) arm in the presence of a dividing wall, and of a testing phase in which the wall was removed and the fish was free to swim for 5 min all over the maze also exploring the novel environment constituted by the N arm. The assignment of circle, square, and triangle to the S, O and N arm was randomized for each experimental subject. The interest in the new environment was evaluated in the testing phase by quantifying the time spent in N arm expressed as percentage of total time, and the number of entries in N arm expressed as percentage of total transitions.

##### Novel Tank Diving Test (NTT)

The apparatus consisted of a glass transparent tank with a lower triangular base of 33- and 30-cm sides and was filled with 4.7 L of water (refer to [49] for graphical detailed description). White paper was used to cover the sides of the tank to avoid reflection. In order to measure vertical exploratory activity, the tank was virtually divided into three equal horizontal areas (bottom, middle and top). The recorded parameters were the number of bottom/middle/top entries, distance travelled within the bottom/middle/top area expressed as percentage of total distance, time spent in the bottom/middle/top area expressed as percentage of total time, and latency to enter the top area expressed as percentage of total time.

##### Light and Dark Preference Test (LDT)

The test was performed in a rectangular plastic tank (L33 × W18 × H18 cm) divided into two compartments of equal size, of which one was made up of opaque white plastic and the other of black plastic. The dark side was shielded from ambient light with an opaque black lid [113] (refer to [49] for graphical detailed description). Before each test, the tank was filled with 4 L of water. In the middle of the tank, two transparent transverse septa restrict the passage between the two areas to prevent the fish from freely swimming from one area to the other. Two transparent sliding doors (W18 × H18 cm) defined a central area (W7 × D18 × H18 cm) that was half black and half white where the zebrafish was housed before starting the test. The experimental subject was placed in the central area and left to settle; then, the two sliding doors were lifted simultaneously to allow the zebrafish to move between the black and white areas for 10 min. Parameters analysed were the time spent in the bright area expressed as percentage of total time and the number of passages between the two areas.

##### Social Preference Test (SPT)

The experimental setting consisted of three rectangular tanks (W28 × D25 × H16 cm), namely empty, experimental, and conspecific tanks aligned side by side in a horizontal line (refer to [49] for graphical detailed description). Each tank was filled with 4 L of water. The experimental tank is in the middle and houses the experimental subject. One of the two adjacent tanks contained three zebrafish of the same size and age as the test subject and represented the social stimulus (conspecific tank); the other tank was left empty. The three fish were novel to the experimental subjects, having never been placed in the same tank together before. The week before the tests, the three fish were transferred to the conspecific tank for 3 h a day to adapt to the tank. The same three fish were used for all social tests. The group position (on the left or the right of the experimental tank) was balanced between tests. At the beginning of the test, two black panels were positioned between the tanks to prevent the experimental subject from seeing the other tanks. The zebrafish were placed in the tank and left to settle, then the two panels were gently removed, and the fish was allowed to swim freely for 10 min. The experimental tank was divided into three virtual areas. The area closest to the conspecific tank was designated as the “social area”, where the fish was assumed to prefer visual interaction with conspecifics. The parameters analysed were the number of entries in the social area, time spent in the social area expressed as percentage of total time, and distance travelled in the social area expressed as percentage of total distance travelled.

##### Mirror Biting Test (MBT)

The apparatus consisted of a barrel tank (W28 × D25 × H16 cm) filled with 4 L of water and equipped with a mirror (D16 × H14 cm). The zebrafish were placed in the tank and left to settle, and the mirror was then tilted at an angle of 22.5° on the long side of the tank [127,130,131] (refer to [49] for graphical detailed description). The position of the mirror (whether the reflected image was closer to the left or right side of the aquarium) was balanced between the tests. The parameters analysed were the number of times crossing the lines denoting the mirror area, the time spent in the mirror area expressed as percentage of total time, number of times that the fish bit the mirror and latency to the first mirror bite expressed as percentage of total time.

## 5. Conclusions

The thermal treatment (34 °C for 21 days) reduced the gene expression of BDNF and its receptors in adult zebrafish, and the reduced or absent expression of BDNF in HT or KO influenced the brain proteome, behaviour and response to thermal variation. Differences in protein expression and behaviour were observed between HT and KO, suggesting that the partial or absent expression of BDNF has a different impact at the neuronal level. A reduction of proteins associated with vesicular trafficking, exocytosis, synapse functioning and neurotransmitted-mediated transduction was observed in KO zebrafish. These neurochemical changes were associated with behavioural alterations that could be interpreted as an increase in zebrafish boldness.

## Figures and Tables

**Figure 1 ijms-23-05606-f001:**
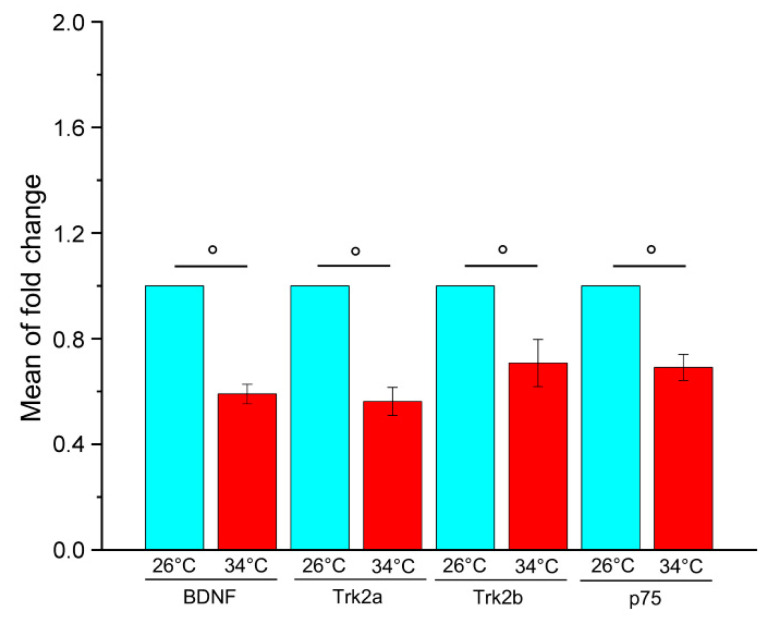
Relative quantification of BDNF, Trk2a, Trk2b and p75 gene expression in the brain of adult WT zebrafish kept at 26 °C and 34 °C for 21 days. For each gene, the expression level at 26 °C was set to 1. The expression levels were normalised against tubulin and expressed as fold change relative to samples at 26 °C. Data are expressed as mean ± SEM and analysed with an unpaired *t*-test, ° *p* < 0.01. N = 12. Blue and red colours refer to 26 °C and 34 °C, respectively. Primer sequences used are reported in Appendix A.

**Figure 2 ijms-23-05606-f002:**
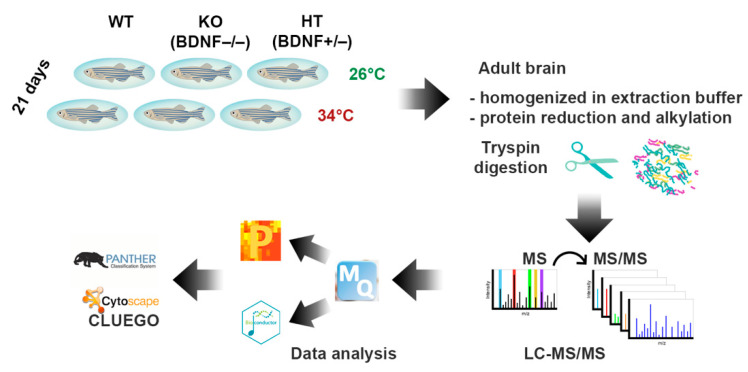
Cartoon of the workflow of the proteomic analysis of the brains of adult zebrafish kept for 21 days at 26 °C and 34 °C.

**Figure 3 ijms-23-05606-f003:**
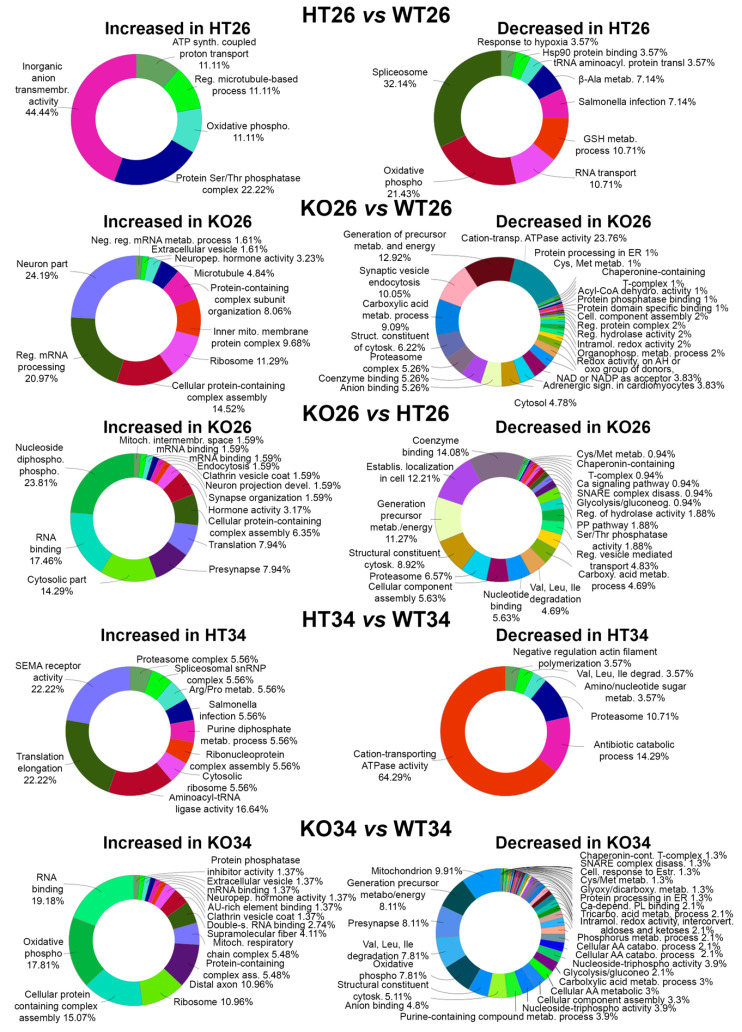
Bioinformatic analysis by ClueGo of the proteins differentially or exclusively expressed in HT26 vs. WT26, KO26 vs. WT26, KO26 vs. HT26, HT34 vs. WT34, KO34 vs. WT34. Bioinformatic analyses were carried out by ClueGo software (Cytoskape release 3.8.2) to cluster enriched annotation groups of biological processes, pathways, and networks within the set of differentially expressed or exclusively expressed proteins in the comparisons. Functional grouping was based on *p* ≤ 0.05. Functional grouping was based on *p* ≤ 0.05 and at least three counts.

**Figure 4 ijms-23-05606-f004:**
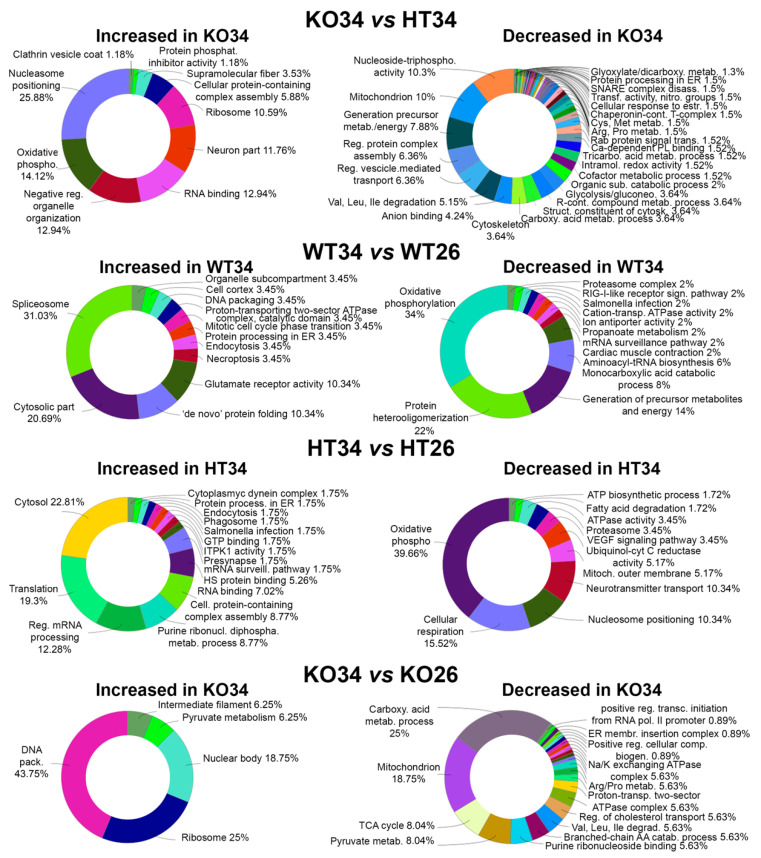
Bioinformatic analysis by ClueGo of the proteins differentially or exclusively expressed in KO34 vs. HT34, WT34 vs. WT26, HT34 vs. HT26, KO34 vs. KO26. Bioinformatic analyses were carried out by ClueGo software (Cytoskape release 3.8.2) to cluster enriched annotation groups of biological processes, pathways, and networks within the set of differentially expressed or exclusively expressed proteins in the comparisons. Functional grouping was based on *p* ≤ 0.05. Functional grouping was based on *p* ≤ 0.05 and at least three counts.

**Figure 5 ijms-23-05606-f005:**
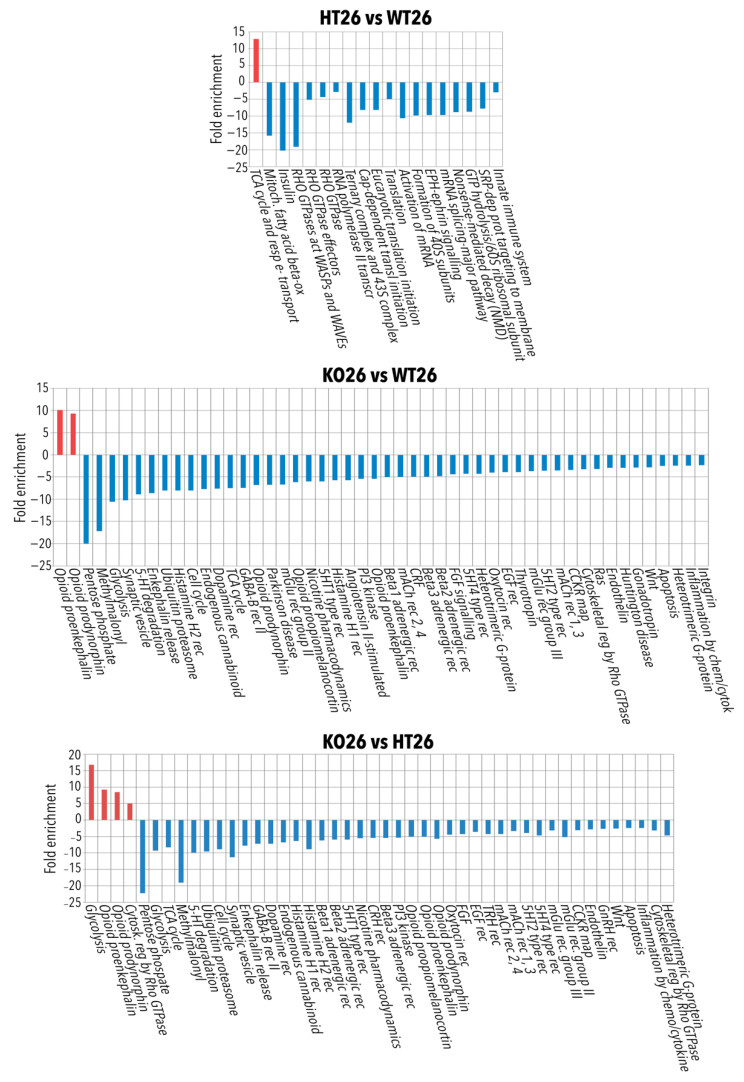
Panther pathways analysis of the proteins differentially or exclusively expressed in HT26 vs. WT26, KO26 vs. WT26 and KO26 vs. HT26. Bioinformatic analyses were conducted by Panther software (release 16.0) to cluster enriched Panther pathway within the set of differentially expressed or exclusively expressed proteins in HT26 vs. WT26, KO26 vs. WT26 and KO26 vs. HT26. If any Panther pathways enrichment was found, the data were processed by Panther Reactome to find Reactome GO and pathways enrichment. Functional grouping was based on Fischer’s exact test *p* ≤ 0.05 and at least three counts.

**Figure 6 ijms-23-05606-f006:**
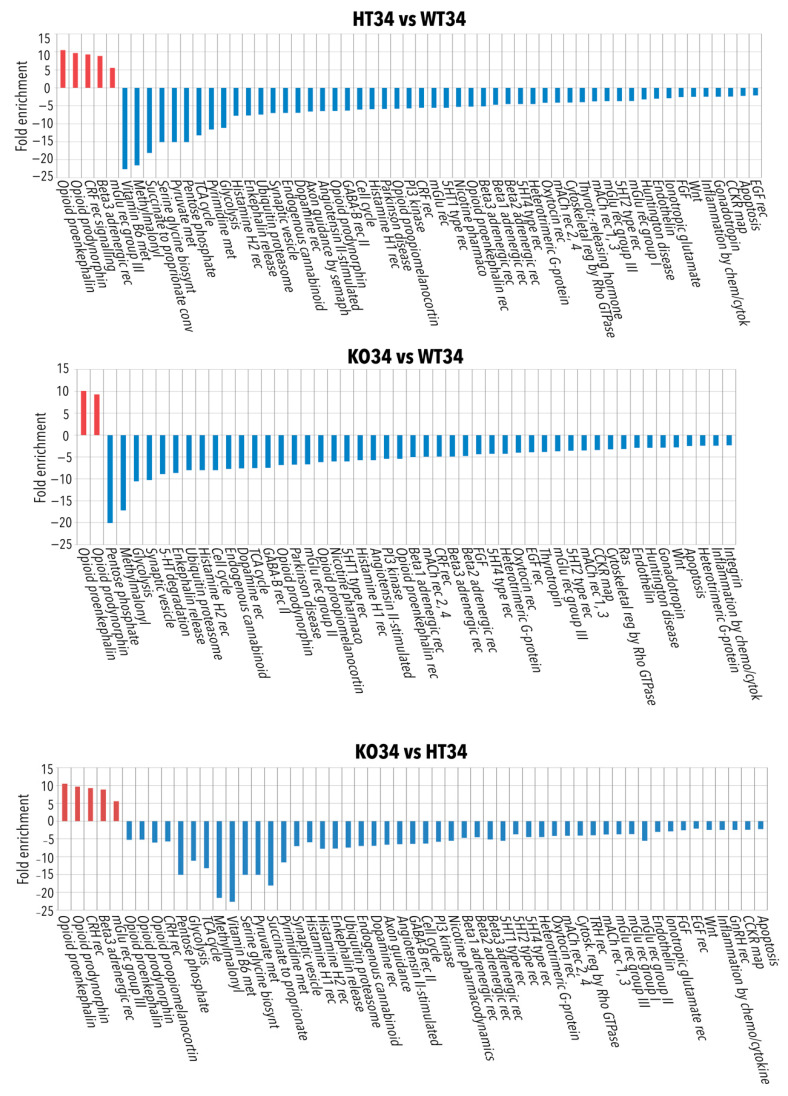
Panther pathways analysis of the proteins differentially or exclusively expressed in HT34 vs. WT34, KO34 vs. WT34 and KO34 vs. HT34. Bioinformatic analyses were conducted by Panther software (release 16.0) to cluster enriched Panther pathway within the set of differentially expressed or exclusively expressed proteins in HT34 vs. WT34, KO34 vs. WT34 and KO34 vs. HT34. If any Panther pathways enrichment was found, the data were processed by Panther Reactome to find Reactome GO and pathways enrichment. Functional grouping was based on Fischer’s exact test *p* ≤ 0.05 and at least three counts.

**Figure 7 ijms-23-05606-f007:**
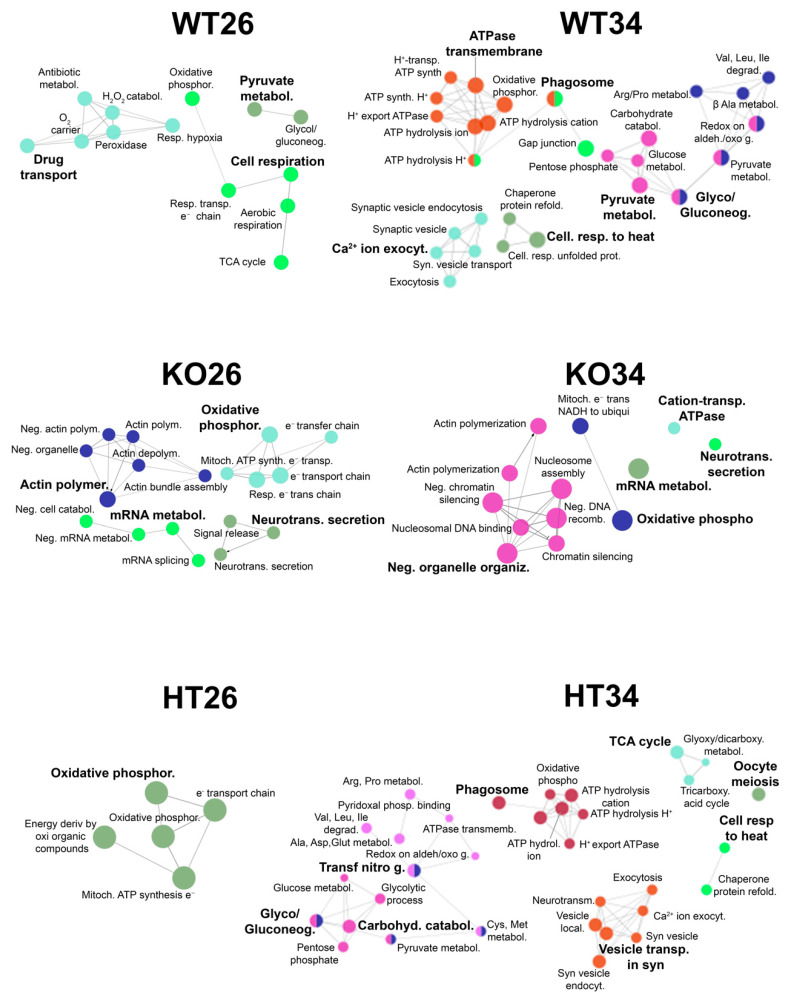
Functional analysis by ClueGo of the proteins most differentiated comparing each data set (WT26, WT34, KO26, KO34, HT26, HT34) with all the others. All data sets were analysed using R-Bioconductor: LFQ data were normalised to the median and analysed for differential expression by the limma package, choosing |Log2FC| > 0.585 and FDR < 0.05 as differential thresholds for the limma test. Each data set was compared with all the others, and the proteins most differentiated were analysed by ClueGo. Functional grouping was based on Fischer’s exact test *p* ≤ 0.05.

**Figure 8 ijms-23-05606-f008:**
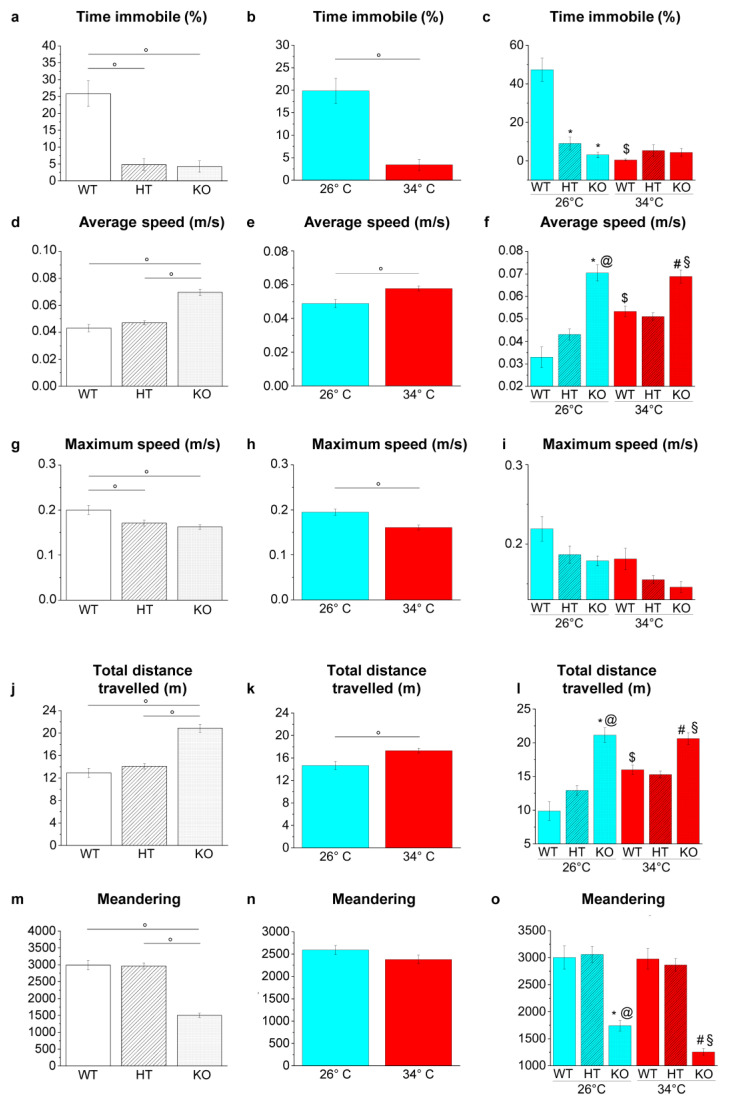
Swimming behaviour and locomotor activities in YMT. (**a**–**c**) Time immobile expressed as percentage of total time; (**d**–**f**) average speed; (**g**–**i**) maximum speed, (**j**–**l**) total distance travelled; (**m**–**o**) meandering. The data are expressed as means ± S.E.M. and analysed by two-way ANOVA with Bonferroni post hoc correction. *p* ≤ 0.05, °; *, HT26 vs. WT26 or KO26 vs. WT26; @, KO26 vs. HT26; #, HT34 vs. WT34 or KO34 vs. WT34; §, KO34 vs. HT34; $, WT34 vs. WT26, HT34 vs. HT26 or KO34 vs. KO26. *p* values and symbols used in the figure are shown in Appendix A. N = 14. Blue and red colours refer to 26 °C and 34 °C, respectively.

**Figure 9 ijms-23-05606-f009:**
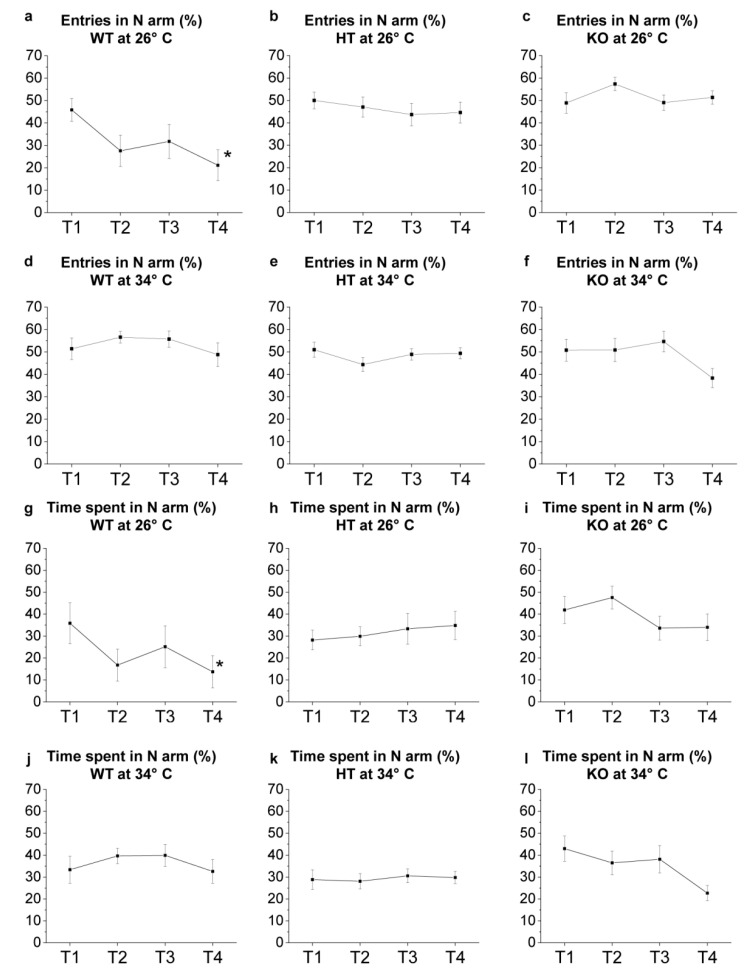
YMT: novel environment exploration T1–T4. (**a**–**f**) Number of entries in N arm expressed as percentage of total entries in Y-maze arms; (**g**–**l**) time spent in N arm expressed as percentage of total time performed by WT (**a**,**d**,**g**,**j**), HT (**b**,**e**,**h**,**k**) and KO (**c**,**f**,**i**,**l**) at 26 °C (**a**–**c**,**g**–**i**) or 34 °C (**d**–**f**,**j**–**l**). The data were analysed by one-way repeated measures ANOVA with post hoc Bonferroni correction. *p* ≤ 0.05, *, compared to T1. *p* values are reported in Appendix A. N = 14.

**Figure 10 ijms-23-05606-f010:**
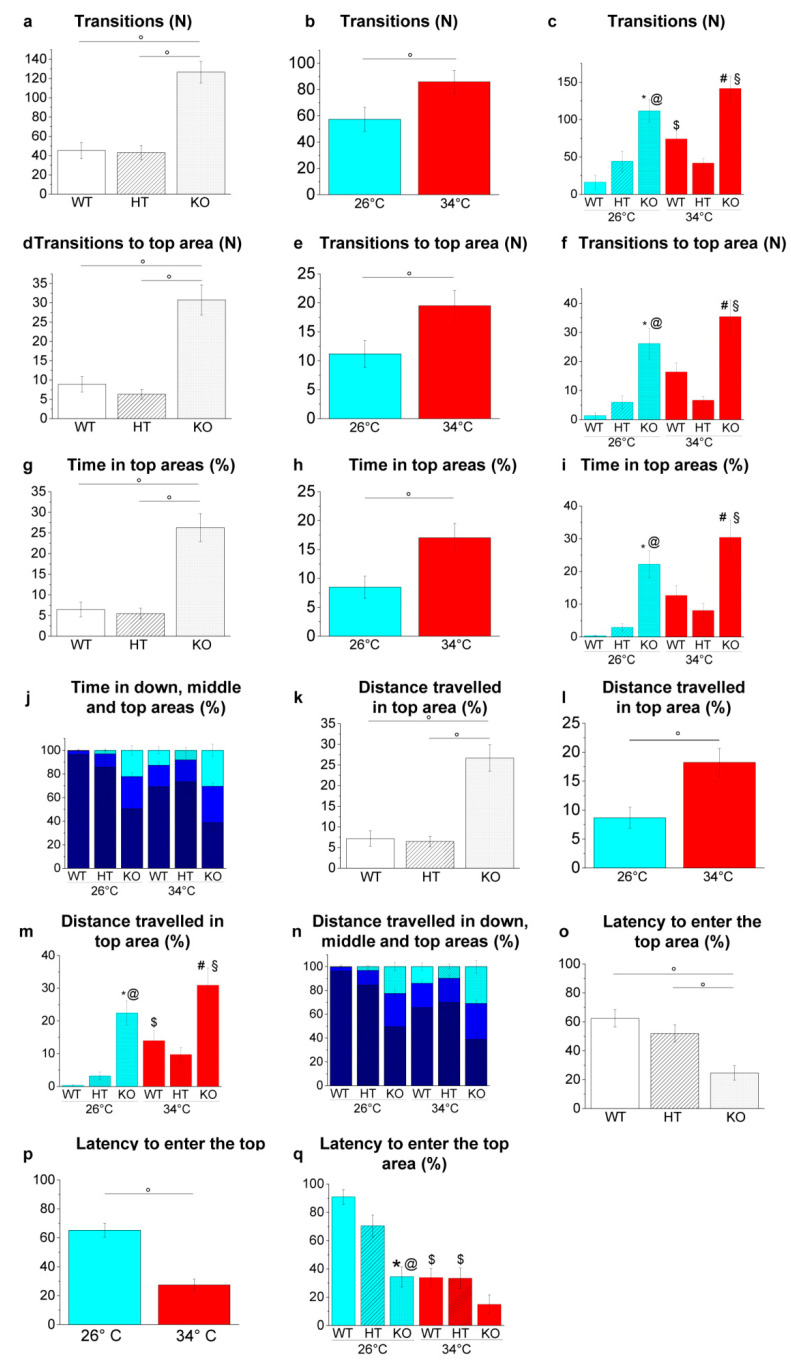
Novel environment exploration behaviour in the NTT. (**a**–**c**) Transitions among areas; (**d**–**f**) transitions to top area; (**g**–**i**) time spent in the top area expressed as a percentage of the total time; (**j**) time spent in the three vertical virtual areas of the tank expressed as a percentage of the total time; (**k**–**m**) distance travelled in the top area expressed as a percentage of the total distance travelled, (**n**) distance travelled in the three vertical virtual areas of the tank expressed as a percentage of the total distance travelled; (**o**–**q**) latency to enter the top area. The data are expressed as means ± S.E.M. and analysed by two-way ANOVA with Bonferroni post hoc correction. *p* ≤ 0.05, °; *, HT26 vs. WT26 or KO26 vs. WT26; @, KO26 vs. HT26; #, HT34 vs. WT34 or KO34 vs. WT34; §, KO34 vs. HT34; $, WT34 vs. WT26, HT34 vs. HT26 or KO34 vs. KO26. *p* values and symbols used in the figure are shown in Appendix A. N = 20. In panels J and N, dark blue, blue and light blue colours refer respectively to bottom, middle and top areas of the tank. In all other panels, blue and red colours refer to 26 °C and 34 °C, respectively.

**Figure 11 ijms-23-05606-f011:**
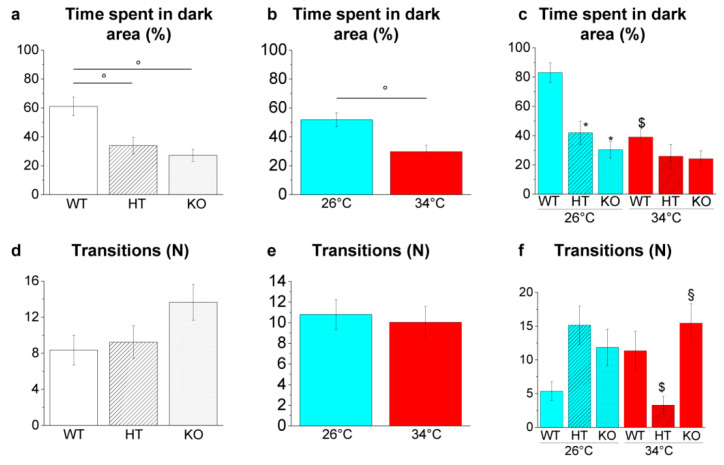
Scototaxis behaviour in the LDT. (**a**–**c**) Time spent in the dark area; (**d**–**f**) number of passages between the bright and dark areas. The data are expressed as mean ± S.E.M. and analysed by two-way ANOVA with Bonferroni post hoc correction. *p* ≤ 0.05, °; *, HT26 vs. WT26 or KO26 vs. WT26; §, KO34 vs. HT34; $, WT34 vs. WT26, HT34 vs. HT26 or KO34 vs. KO26. *p* values and symbols used in the figure are shown in Appendix A. N = 20. Blue and red colours refer to 26 °C and 34 °C, respectively.

**Figure 12 ijms-23-05606-f012:**
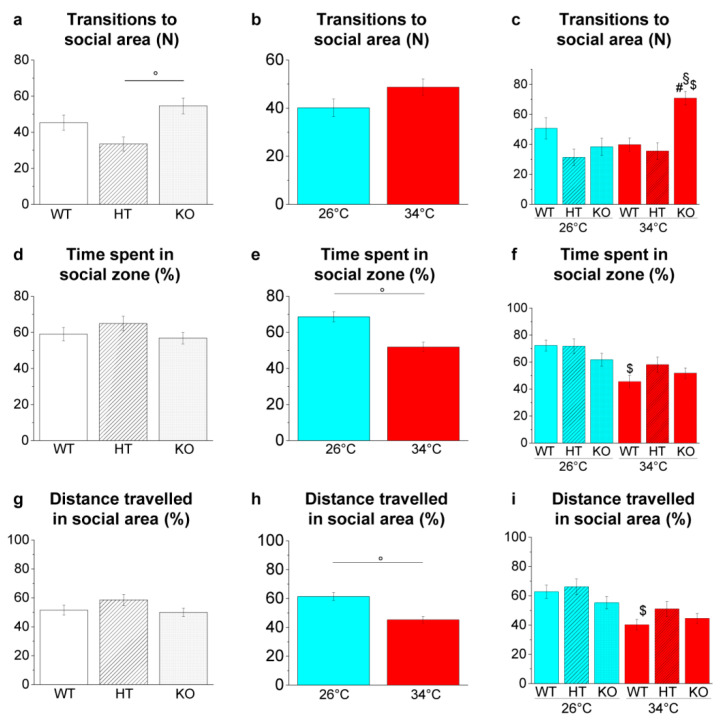
Shoaling behaviour in the SPT. (**a**–**c**) Number of entries in the social area; (**d**–**f**) time spent in the social area expressed as percentage of total time; (**g**–**i**) total distance travelled in the social area expressed as percentage of total time. The data are expressed as mean ± S.E.M. and analysed by two-way ANOVA with Bonferroni post hoc correction. *p* ≤ 0.05, °; #, HT34 vs. WT34 or KO34 vs. WT34; §, KO34 vs. HT34; $, WT34 vs. WT26, HT34 vs. HT26 or KO34 vs. KO26. *p* values and symbols used in the figure are shown in Appendix A. N = 20. Blue and red colours refer to 26 °C and 34 °C, respectively.

**Figure 13 ijms-23-05606-f013:**
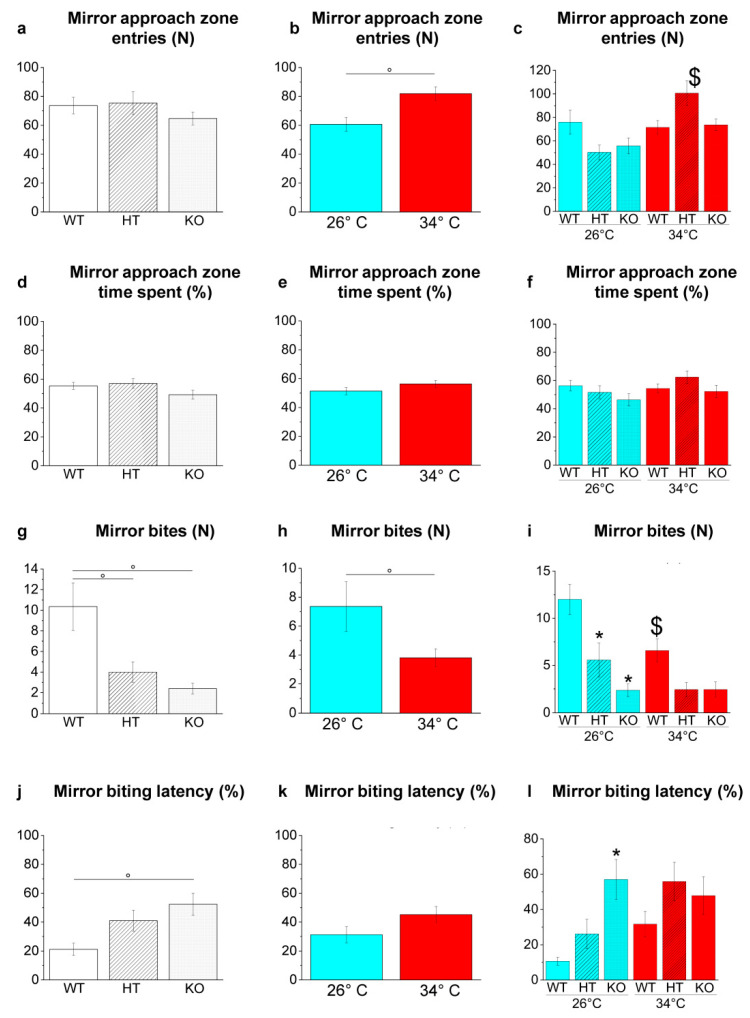
Aggressive behaviour in the MBT. (**a**–**c**) Entries in mirror approach area; (**d**–**f**) time spent in mirror approach area expressed as percentage of total time; (**g**–**i**) mirror bites; (**j**–**l**) mirror-biting latency. The data are expressed as mean ± S.E.M. and analysed by two-way ANOVA with Bonferroni post hoc correction. *p* ≤ 0.05, °; *, HT26 vs. WT26 or KO26 vs. WT26; $, WT34 vs. WT26, HT34 vs. HT26 or KO34 vs. KO26. *p* values and symbols used in the figure are shown in Appendix A. N = 14. Blue and red colours refer to 26 °C and 34 °C, respectively.

**Figure 14 ijms-23-05606-f014:**
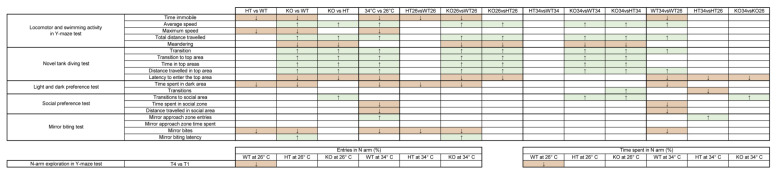
Schematic summary of behavioural analysis reported in Appendix A and in Figure 8, Figure 9, Figure 10, Figure 11 and Figure 12. HT26 vs. WT26, KO26 vs. WT26, KO26 vs. HT26, HT34 vs. WT34, KO34 vs. WT34, KO34 vs. HT34, WT34 vs. WT26, HT34 vs. HT26 and KO34 vs. KO26 are considered. Reduction ↓, increase ↑.

## Data Availability

The mass spectrometry proteomics data have been deposited to the Proteo-meXchange Consorti-um via the PRIDE [126] partner repository, with the dataset identifier PXD030733.

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
