# Peer review of "Brain Proteome and Behavioural Analysis in Wild Type, BDNF+/− and BDNF−/− Adult Zebrafish (Danio rerio) Exposed to Two Different Temperatures"

_ijms, 2022, doi:10.3390/ijms23105606_

Round 1

Reviewer 1 Report

The role of BDNF in behavior and brain (processes) was studied on zebrafish model organisms. Wild type, homozygous and knockout BDNF zebrafish were kept in normal and increased temperature. Behavioral analysis and proteomic analysis of brains were used to elucidate the influence of genotype and temperature.
Authors used appropriate methods, results are clearly presented and conclusions are supported by results.

Author Response

Response to the scientific Reviewer 1’s comments

Manuscript title: Brain proteome and behavioural analysis in wild type, BDNF+/- and BDNF-/- adult zebrafish (Danio rerio) exposed to two different temperatures (Manuscript ID: ijms-1658523)

Rev: the role of BDNF in behavior and brain (processes) was studied on zebrafish model organisms. Wild type, homozygous and knockout BDNF zebrafish were kept in normal and increased temperature. Behavioral analysis and proteomic analysis of brains were used to elucidate the influence of genotype and temperature. Authors used appropriate methods, results are clearly presented and conclusions are supported by results.

Au: we want to thank the scientific Reviewer and the Editor for giving us the opportunity to submit a revised draft of our manuscript for publication in the International Journal of Molecular Sciences.

Reviewer 2 Report

In this study Authors have highlighted importance of BDNF expression in the nervous system of adult zebrafish under control and heat treatment conditions. While the study has highlighted some diverse proteomic and behavioral changes, I have some additional comments/questions before I can give my recommendations for the editorial decision of this manuscript:

1) While I appreciate the amount of diverse data that authors have shown in the manuscript, it will be helpful for the readers if authors include some important highlights and conclusions of the overall results within main text and abstract.

2) Authors state in Fig 1 that expression levels were normalized against tubulin. Can authors please explain why they chose this particular method of normalization. Also it will be interesting to see  the protein levels of tubulin from the mass spectrometer proteomics expression.

3) How was the mass spectrometer data normalized. Can authors add the final proteomics output data used for statistical/pathway enrichment analysis. 

Author Response

Response to the scientific Reviewer 2’s comments

Manuscript title: Brain proteome and behavioural analysis in wild type, BDNF+/- and BDNF-/- adult zebrafish (Danio rerio) exposed to two different temperatures (Manuscript ID: ijms-1658523)

Rev: in this study Authors have highlighted importance of BDNF expression in the nervous system of adult zebrafish under control and heat treatment conditions. While the study has highlighted some diverse proteomic and behavioral changes, I have some additional comments/questions before I can give my recommendations for the editorial decision of this manuscript:

While I appreciate the amount of diverse data that authors have shown in the manuscript, it will be helpful for the readers if authors include some important highlights and conclusions of the overall results within main text and abstract.

AU: as suggested by the Reviewer, we have included the highlights in the abstract and modified the Conclusion section.

Rev: authors state in Fig 1 that expression levels were normalized against tubulin. Can authors please explain why they chose this particular method of normalization. Also it will be interesting to see the protein levels of tubulin from the mass spectrometer proteomics expression.

AU: we performed preliminary experiments to test the expression of four housekeeping genes (β-actin, elongation factor 1α, ribosomal protein L13 and α-tubulin) at 26° C and 34° C to verify the stability of their expression at the two temperatures. All housekeeping genes showed good stability and we chose tubulin for the higher reaction efficiency (lowest CT values) as shown in the new Figure S2. We have added this information to the Materials and Methods section. Regarding the protein expression of tubulin,  the label-free proteomic approach applied in the present study allows only a relative quantification of the proteins present in  each comparison and is not suitable for providing  a general quantification of the  level of  protein expression.

Rev: how was the mass spectrometer data normalized. Can authors add the final proteomics output data used for statistical/pathway enrichment analysis.

AU: The proteomic data were normalized by the Max Quant software based on the LFQ (Label free quantitation) algorithms according to (Cox J, Hein M, Luber C, Paron I, Nagarai N and Mann M. Accurate Proteome-wide Label-free Quantification by Delayed Normalization and Maximal Peptide Ratio Extraction, Termed MaxLFQ. Mol Cell Proteomics. 13(9): 2513–2526, 2014). This information is now provided in the Materials and Methods section. The proteomic data used for the enrichment analysis are the proteins differentially or exclusively expressed in the various comparisons that are listed in the Supplementary Tables S4-S12.

All changes made are tracked by the Microsoft Word “Track Changes” function.

We want to thank the scientific Reviewer and the Editor for the accurate work that contributed to improve our paper and hope that it may now be accepted for publication in the International Journal of Molecular Sciences.

Reviewer 3 Report

In their paper Maffioli et al investigated the response to heat in different BDNF genotypes at the level of the proteome and by different behaviour assays. Overall the paper is technically sound and the topic of interest. The study has clear merit, is well written and it of interest for the reader of IJMS. The authors aim at explaining changes in behaviour based on the proteomics data they generated, but such integrated analysis fall short, due to the current way the proteomics data are analysed. So far the results on the proteomics are not following the current gold standard approaches, which impair the interpretation of the molecular effects and how they can translate to behaviour effects. The authors should go in an indepth integrative analysis of heat and genotype in order to understand how they can change behaviour. Below are some points, that I hope will help the authors to further improve their manuscript.

For the interpretation of the proteomics data the authors relied solely on pairwise differential analysis and pathway analysis using Cytoscape. Given the complexity of the data (3 genotypes and 2 temperature), such analysis (if valid when 1 condition is compared to the other) is insufficient to analyse the combined effects of both heat and genotypes. In this regard the proteomics analysis appears in the current state simplistic, partial and preliminary. I advise the authors to do a joint analysis (for instance by clustering of the normalized expression data), in order to better analyse the effects of heat and genotype and decipher the protein and pathways deregulated by a single or both factors. To do so the authors must apply the current gold standard methods used for proteomics data analysis, see for instance DOI: 10.1093/bib/bbx031 and DOI: 10.1021/acs.jproteome.6b01050. This will ensure that the proteomics data analyses are analysed following the current gold standard in this field and allow the authors to assess how heat or genotype could contribute to the change in behaviour. I strongly advise the authors to contact a bioinformatics team if needed, as this will ensure that the analysis is properly done.

The same is true for the figures on pathway analysis which depict one by one the different enriched term from the pair wise comparison. The comparison of the deregulated pathway in respect to the genotype and/or the heat is currently not possible. The data generated by the authors being complex, a more integrated view of the deregulated pathway is required. In this respect many tools exist that allow to visualize the deregulated pathways in common or unique to several conditions (see for instance the R package ClusterProfiler). I strongly advise the authors to go beyond the pairwise analysis, as the integrated analysis will allow them to identify the pathways responding to either heat or BNDF genotype, or both, and thus conclude more convincingly on the molecular effects and how they can explain the change in behaviour.

Author Response

Rev: In their paper Maffioli et al investigated the response to heat in different BDNF genotypes at the level of the proteome and by different behaviour assays. Overall the paper is technically sound and the topic of interest. The study has clear merit, is well written and it of interest for the reader of IJMS. The authors aim at explaining changes in behaviour based on the proteomics data they generated, but such integrated analysis fall short, due to the current way the proteomics data are analysed. So far the results on the proteomics are not following the current gold For the interpretation of the proteomics data the authors relied solely on pairwise differential analysis and pathway analysis using Cytoscape. Given the complexity of the data (3 genotypes and 2 temperature), such analysis (if valid when 1 condition is compared to the other) is insufficient to analyse the combined effects of both heat and genotypes. In this regard the proteomics analysis appears in the current state simplistic, partial and preliminary. I advise the authors to do a joint analysis (for instance by clustering of the normalized expression data), in order to better analyse the effects of heat and genotype and decipher the protein and pathways deregulated by a single or both factors. To do so the authors must apply the current gold standard methods used for proteomics data analysis, see for instance DOI: 10.1093/bib/bbx031 and DOI: 10.1021/acs.jproteome.6b01050. This will ensure that the proteomics data analyses are analysed following the current gold standard in this field and allow the authors to assess how heat or genotype could contribute to the change in behaviour. I strongly advise the authors to contact a bioinformatics team if needed, as this will ensure that the analysis is properly done. The same is true for the figures on pathway analysis which depict one by one the different enriched term from the pair wise comparison. The comparison of the deregulated pathway in respect to the genotype and/or the heat is currently not possible. The data generated by the authors being complex, a more integrated view of the deregulated pathway is required. In this respect many tools exist that allow to visualize the deregulated pathways in common or unique to several conditions (see for instance the R package ClusterProfiler). I strongly advise the authors to go beyond the pairwise analysis, as the integrated analysis will allow them to identify the pathways responding to either heat or BNDF genotype, or both, and thus conclude more convincingly on the molecular effects and how they can explain the change in behaviour.

AU: we thank the Reviewer for the suggestion, however one of the aims of the present study was to investigate how the variation in BNDF expression possibly altered the response to temperature change described in our previous studies on the subject (Toni M  et al.(2019) Environmental temperature variation affects brain protein expression and cognitive abilities in adult zebrafish (Danio rerio): A proteomic and behavioural study Journal of Proteomics, Vol. 204:103396, Nonnis et al., Impact of acute thermal stress on Zebrafish brain, Sci Rep. 2021 28;11(1):252); therefore we adopted the same proteomic protocols in order to compare results obtained under the same experimental conditions and using the same post-acquisition data analysis approach. We better clarified this aspect in the text.

All changes made are tracked by the Microsoft Word “Track Changes” function.

We want to thank the scientific Reviewer and the Editor for the accurate work that contributed to improve our paper and hope that it may now be accepted for publication in the International Journal of Molecular Sciences.

Round 2

Reviewer 2 Report

Authors have addressed all my comments by adding additional information in the manuscript. I recommend this manuscript for publication in present form. 

Author Response

Reviewer 2

Authors have addressed all my comments by adding additional information in the manuscript. I recommend this manuscript for publication in present form.

AU: We thank again the Reviewer for the helpful advice that enabled us to improve the quality of the manuscript. The revised version of the manuscript includes the additions made following the comments of the other scientific reviewers.

Reviewer 3 Report

The authors provided a novel version of their manuscript. They want to use the same methodology in order to compare the effect avec BDNF genotype on the response to heat. I do not advise to do so as this can lead to misinterpretation of their data. I advise them to answer to the points below.

In the discussion section the authors state:

“The comparison of KO34vsKO26 and WT34vsWT26 allowed us to evaluate whether thermal increase determines a different alteration in subjects that do not express BDNF and in those that express the neurotrophin. The high temperature in KO caused a reduction in proteins related to amino acid metabolism, glycolysis, mitochondrion, cell junctions, vesicular transport, phagosome, and signal transduction pathways that was not observed in WT. This suggests that the lack of BDNF alters the cellular response to heat treatment, potentially making the nervous system more vulnerable to changes in the external environment.”

With this sentence, the authors state that they provided results on how the response to heat is changed when BDNF is mutated or not. The problem is that the authors can not conclude such thing from the pairwise comparisons the show. Indeed, KO34 vs KO26 will provide a list of biological process that are deregulated by heat in the context of BDNFKO. On the other side WT 34 vs WT26 will provide the response to heat in the context of WT situation. But it is wrong to compare directly these pathways, as protein deregulated in common in both comparative analysis can bias the pathway enrichment. To conclude on how BDNF genotype influence the heat response the authors must first substract the protein regulated in common in KO34 vs KO26 and WT 34 vs WT26, and only then proceed to pathway enrichment analysis. The figure 7 can not be used for such argument as, protein deregulated in common in the KO34 vs KO26 and WT 34 vs WT26 will contribute the enrichment of the pathways in both genotype, while they can not explain the response to heat when BDNF is mutated or not.

This whole figure 7 can be misleading, as the authors can not delineate the respective contribution of the deregulation due to heat or genotype. This is for me a clear limitation of the pairwise analysis.

What are the proteins in common or different in the WT, HT and KO at the different temperature? To which biological processes they belong to? Answers to these basic questions are not supported by results in this version of their paper.

I do not understand the choice of the authors to keep the analysis under this form. The argument of comparing to older data do not stand, as the point of this paper is to analyse to contribution of the BDNF genotype. I advise the authors, to consider replacing the figure 7 by a proper analysis of the normalized protein counts by limma, followed by hierarchical clustering and enrichment analysis on each cluster. This is the only reasonable way to analyse their complex dataset. And will greatly improve the quality of their manuscript.

Author Response

AU: as suggested by the Reviewer, to better discriminate the effects of heat and genotype the data sets were further analysed using R-Bioconductor by Dr. Ivan Arisi, a bioinformatic researcher who is added as co Author of the manuscript. In detail: LFQ data were normalized to median and analysed for differential expression by the limma package, choosing |Log2FC|>0.585 and FDR<0.05 as differential thresholds for the limma test. The differential analysis was followed by hierarchical clustering and heatmap of samples and genes. We replace Fig. 7 with a new figure reporting the enrichment analysis carried out upon clusterization. The text of the manuscript has been modified and integrated with the information deriving from the analysis carried out with limma. We thank the Reviewer for valuable advice which we believe has resulted in an improvement in the quality of the manuscript.

Round 3

Reviewer 3 Report

The authors answered correctly to my comments and suggestions, and support the publication of their manuscript in IJMS